# *From Chaos to Clarity*: Claim Normalization to Empower Fact-Checking

**Megha Sundriyal**[1,3]**, Tanmoy Chakraborty**[2]**, Preslav Nakov**[3]
[1]Indraprastha Institute of Information Technology Delhi, India
[2]Indian Institute of Technology Delhi, India
[3]Mohammed Bin Zayed University of Artificial Intelligence
meghas@iiitd.ac.in, tanchak@iitd.ac.in, preslav.nakov@mbzuai.ac.ae

## Abstract

With the rise of social media, users are exposed to many misleading claims. However, the pervasive noise inherent in these posts presents a challenge in identifying precise and prominent claims that require verification. Extracting the important claims from such posts is arduous and time-consuming, yet it is an underexplored problem. Here, we aim to bridge this gap. We introduce a novel task, *Claim Normalization* (*aka ClaimNorm*), which aims to decompose complex and noisy social media posts into more straightforward and understandable forms, termed *normalized claims*. We propose CACN, a pioneering approach that leverages chain-of-thought and claim check-worthiness estimation, mimicking human reasoning processes, to comprehend intricate claims. Moreover, we capitalize on the in-context learning capabilities of large language models to provide guidance and to improve claim normalization. To evaluate the effectiveness of our proposed model, we meticulously compile a comprehensive real-world dataset, CLAN, comprising more than $6k$ instances of social media posts alongside their respective normalized claims. Our experiments demonstrate that CACN outperforms several baselines across various evaluation measures. Finally, our rigorous error analysis validates CACN's capabilities and pitfalls.[1]

## 1 Introduction

Social media have enabled a new way of communication, breaking down geographical barriers and bringing unprecedented opportunities for knowledge exchange. However, this has also presented a growing threat to society, e.g., during the 2016 US Presidential Election (Allcott and Gentzkow, 2017), the COVID-19 pandemic (Alam et al., 2021; Rocha et al., 2021; Nakov et al., 2022a), the Ukraine–Russia conflict (Khaldarova and Pantti, 2016), etc.

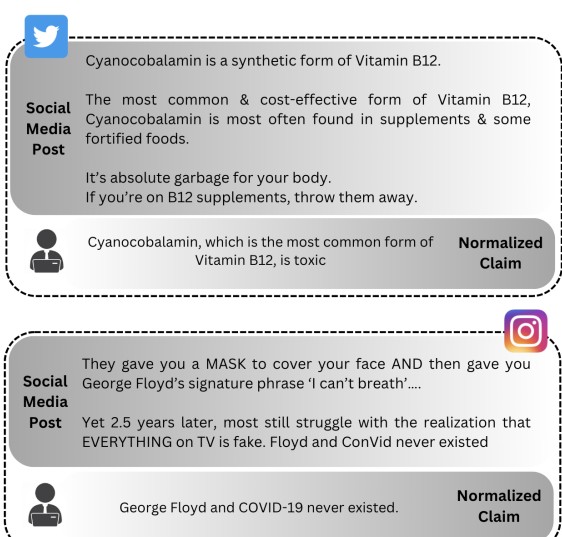

Figure 1: Illustration of our proposed *Claim Normalization* task, highlighting the normalized claims authored by fact-checkers for social media posts from distinct social media platforms.

False claims are an intrinsic aspect of fabricated news, rumors, propaganda, and misinformation. Journalists and fact-checkers work tirelessly to assess the factuality of such claims in spoken and/or written form, sifting through an avalanche of claims and pieces of evidence to determine the truth. To further address this pressing issue, several independent fact-checking organizations have emerged in recent years, such as Snopes,[2] FullFact,[3] and PolitiFact,[4] which play a crucial role in verifying the accuracy of online content. However, the rate at which online information is being disseminated far outpaces the capacity of fact-checkers, making it difficult to verify every single claim. This, in turn, leaves numerous unverified claims circulating online, potentially reaching millions before they can be verified.

---

[1]We release our dataset and code at 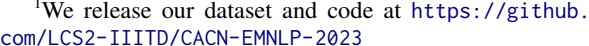 https://github.com/LCS2-IIITD/CACN-EMNLP-2023

[2]https://www.snopes.com
[3]https://fullfact.org
[4]https://www.politifact.com

While the complete automation of the fact-checking pipeline may pose hazards to accountability and reliability, several recent studies have targeted identifying downstream tasks suitable for automation, such as detecting claims (Daxenberger et al., 2017; Chakrabarty et al., 2019; Gangi Reddy et al., 2022b), evaluating their worthiness for fact-checking (Gencheva et al., 2017; Jaradat et al., 2018; Wright and Augenstein, 2020), making sure they were not fact-checked before (Shaar et al., 2020, 2022a,b; Hardalov et al., 2022), and validating them by retrieving relevant shreds of evidence (Zhi et al., 2017; Hanselowski et al., 2018; Soleimani et al., 2020; Pan et al., 2023). See also a recent survey on automated fact-checking for assisting human fact-checkers (Nakov et al., 2021a).

In light of the growing challenges faced by fact-checkers in verifying the factuality of social media claims, we propose the novel task of *claim normalization*. This task aims to extract and to simplify the central assertion made in a long, noisy social media post. This can improve the efficacy and curtail the workload of fact-checkers while maintaining high precision and conscientiousness. We provide a more detailed explanation of why the claim normalization task is essential and illustrate its significance in Appendix A.1.

In our problem formulation, given an input social media claim, the system needs to simplify it in a concise form that contains the post's central assertion that fact-checkers can easily verify. To better understand our motivation, we illustrate the task in Figure 1. The first social media post reads, '*Cyanocobalamin is a synthetic form of Vitamin B12...If you're on B12 supplements, throw them away.*' This post contains some extraneous information that has no relevance for fact-checkers. As a result, they distil the information and summarize it as, '*Cyanocobalamin, the most common form of Vitamin B12, is toxic.*' Fact-checkers tasked with verifying the accuracy of such noisy posts need to read through them and condense their content to obtain a concise claim that can be easily fact-checked. Unfortunately, this process can be exceedingly time-consuming. By automating the claim normalization process, fact-checkers can work more efficiently. Another aspect is that fact-checkers often choose what to fact-check based on the virality of a claim, for which they need to be able to recognize when the same claim appears in a slightly different form, and claim normalization is essential for this.

Our contributions are as follows:

- We introduce the novel task of *claim normalization*, which seeks to detect the core claim in a given piece of text.

- We present a meticulously curated high-quality dataset specifically tailored for claim normalization of noisy social media posts.

- We propose a robust framework for claim normalization, incorporating chain-of-thought, in-context learning, and claim check-worthiness estimation to comprehend intricate claims.

- We conduct a thorough error analysis, which can inform future research.

## 2 Related Work

**Claim Analysis.** Previous work has focused on distinct aspects of claims, including claim detection (Daxenberger et al., 2017; Gupta et al., 2021; Sundriyal et al., 2021; Gangi Reddy et al., 2022a,b), claim check-worthiness estimation (Hassan et al., 2017; Gencheva et al., 2017; Barrón-Cedeño et al., 2018; Jaradat et al., 2018; Vasileva et al., 2019; Barrón-Cedeño et al., 2020; Konstantinovskiy et al., 2021), claim span identification (Sundriyal et al., 2022), etc. By curating the AAWD corpus, Bender et al. (2011) pioneered the efforts in claim detection, the foremost step in the fact-checking tasks. Following this, linguistically motivated features, including sentiment, syntax, context-free grammar, and parse trees, were frequently used (Daxenberger et al., 2017; Lippi and Torroni, 2015; Levy et al., 2017; Sundriyal et al., 2021). Recently, large language models (LLMs) have also been used for claim detection (Chakrabarty et al., 2019; Barrón-Cedeño et al., 2020; Gupta et al., 2021; Gangi Reddy et al., 2022a,b).

Most previous work on claim detection and extraction primarily concentrated on adapting to text that comes from similar distributions or topics. Moreover, it often relied on well-structured formal writing. In contrast, our objective is to develop a system that specifically addresses the challenges posed by posts in social media and aims to extract the central claim in a more simplified manner, which goes beyond extracting a text subspan in a social media post and aims at abstractive claim extraction that *mimics what professional fact-checkers do*. To the best of our knowledge, we are the first to address the task of claim extraction in this very practical formulation.

**Text Summarization.** The task of claim normalization is closely related to the task of text summarization. In the latter, given a lengthy document, the goal is to summarize it into a much shorter summary. Previous work on text summarization has explored various approaches, including large pre-trained seq2seq models to generate high-quality summaries (Radford et al., 2019; Lewis et al., 2020; Raffel et al., 2020).

One issue has been the faithfulness of the summary with respect to the source. To address this, Kryscinski et al. (2020) introduced FactCC, a weakly-supervised BERT-based entailment model, which augments the dataset with artificially introduced faithfulness errors. Similarly, (Utama et al., 2022) trained a model for detecting factual inconsistencies in data from controllable text generation that perturbs human-annotated summaries, introducing varying types of factual inconsistencies. Durmus et al. (2020) proposed a question-answering framework that compares answers from the summary to those from the original text.

All these approaches primarily focused on general-purpose summarization and did not provide means for models to generate summaries primarily focusing on specific needs. To address this limitation, controlled summarization was introduced (Fan et al., 2018). One aspect of controlled summarization is length control, in which users can set their preferred summary length (Rush et al., 2015; Kikuchi et al., 2016). Recent research has discovered that, despite their fluency and coherence, state-of-the-art abstractive summarization systems produce summaries with contradictory information.

While text summarization systems can assist in condensing social media posts into shorter summaries, their primary goal is not to ensure verifiability. It aims to capture the key points of the text rather than emphasizing the specific claims within the text that need to be fact-checked. Our task of claim normalization, on the other hand, works at an entirely different level. It needs a thorough understanding of the claims made in the social media post and strives to ensure that the normalized claims are not only consistent with the original post, but are also self-contained and verifiable.

Despite the progress in text summarization, the task of claim normalization remains underexplored. In this work, we aim to tackle this challenging problem by developing a robust approach specifically tailored to the unique aspects of this task.

# 3 Dataset

Existing text summarization datasets have not specifically addressed the need for claim-oriented summaries. To address this gap, we propose a novel dataset CLAN (**Cla**im **N**ormalization), consisting of fact-checked social media posts paired with concise claim-oriented summaries (known as *normalized claims*), created by fact-checkers as part of the verification process. As a result, our dataset is not subjected to external annotation, thus averting potential biases and ensuring its high quality.

## 3.1 Data Collection

We gathered our fact-checked post and claim pairs from two sources: (*i*) Google Fact-Check Explorer[5] and (*ii*) ClaimReview Schema.[6]

**Google Fact-Check Explorer.** We acquired a list of fact-checked claims from multiple reputed fact-check sources via Google Fact-Check Explorer's API (GFC). This data collection pipeline followed a three-step process. First, we extracted the *title*, which is usually a single-sentence short summary of the information being fact-checked, and the *fact-checking site's URL*. This step yielded a total of 22,405 unique fact-checks. We then proceeded to retrieve the social media post and the associated *claim review* if they were available on the fact-checking site. Due to the collected posts having already undergone fact-checking and containing misleading claims, a significant number of them were unavailable for inclusion in our dataset. Moreover, a significant number of the posts only contained images or videos, which were unsuitable for our task at hand. As a result, we were left with a considerably smaller number of relevant instances. We also noted that in certain instances, the *title* in the Google Fact-Check Explorer and the *claim review* were identical; consequently, we included only one in the final dataset.

**The ClaimReview Schema.** We targeted the ClaimReview Schema elements with an entry for *reviewed items* as they were relevant to our requirements. Out of 44,478 entries, only 22,428 had this particular field. Therefore, we had to filter out the remaining entries. Next, we extracted all the links to social media posts and their corresponding claim reviews provided by the fact-checkers.

---

[5]https://toolbox.google.com/factcheck/explorer
[6]https://schema.org/ClaimReview

| | Social Media Post | Normalized Claim |
|---|---|---|
| 1 | Research into the dangers of cooking with aluminum foil has found that some of the toxic metal can contaminate food. Increased levels of aluminum in the body have been linked to osteoporosis, and Alzheimer's disease. | Cooking in Aluminum foil causes Alzheimer's Disease. |
| 2 | Did you know when ur child turns 6. U can add them as authorized user to one of ur credit cards. Never give them card, & all payments u make from 6 to 18 goes to ur child credit too..ur kid will have a unbelievable credit score from years of payment history. | 6-year-old kids can be added as authorized users on all credit cards. |
| 3 | As if it couldn't get any worse. #Hope4Cancer says #RootCanal causes #CANCER Solution... Rip Cancer patients teeth out. Monsters #FalseHope4Cancer #ProtectCancerPatients | Having a root canal can cause cancer. 

 Root canal treatment causes cancer |

Table 1: Examples of social media posts and their corresponding normalized claims from CLAN. The first two examples come from the training set and each has one reference normalized claim, while the last one comes from the test set, and thus it has two reference normalized claims.

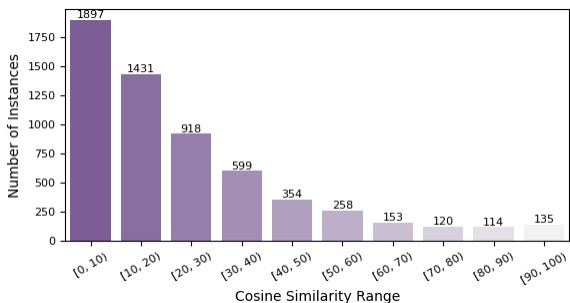

Figure 2: Histogram of the cosine similarity between the social media posts and the corresponding normalized claims from our CLAN dataset.

As mentioned above, we only processed textual claims and excluded other modalities, such as audio or video. Further, we ensured that all the entries were in English.

## 3.2 Data Statistics and Analysis

By using both of these data collection methods and by exercising careful consideration, we curated a total of 6,388 instances. To ensure the creation of a diverse and high-quality test set, we chose posts that comprised not one, but two reference normalized claims (*c.f.* Sec 3.1). This enabled us to capture different aspects and perspectives of the normalized claims by including multiple references, thereby increasing the test set's robustness and reliability. Representative examples from our dataset are shown in Table 1, and the final dataset statistics are shown in Table 2.

| Dataset | Train | Val | Test | Overall |
|---|---|---|---|---|
| Total number of pairs | 5,341 | 594 | 453 | 6,388 |
| Avg. length of posts | 39.52 | 37.12 | 57.97 | 44.87 |
| Avg. length of claims | 16.47 | 17.24 | 15.41 | 16.37 |

Table 2: Statistics about our CLAN dataset.

Figure 2 shows an analysis of the cosine similarities between the social media posts and the corresponding normalized claims. We can see that the cosine similarities are consistently low for most examples, demonstrating that claim normalization involves more than just summarizing the social media post. This highlights the need for a specialized effort to accurately identify, extract, and normalize the claims within social media posts.

## 4 Proposed Approach

In this section, we explain our proposed approach, **C**heck-worthiness **A**ware **C**laim **N**ormalization (CACN), which aims to integrate task-specific information with large language models (LLMs). We focus our experiments on GPT-3 (text-davinci-003) (Brown et al., 2020). Our approach amalgamates two key ideas: (*i*) chain-of-thought prompting and (*ii*) reverse check-worthiness.

**Chain-of-Thought Prompting.** The realm of chain-of-thought (CoT) prompting has emerged as a veritable tour de force within LLMs (Wei et al., 2022). Instead of undergoing the laborious process of fine-tuning individual model checkpoints for every new task, we use CoT to navigate the complexity of claim normalization by using step-by-step reasoning. To accomplish this, we use claim check-worthiness, as described in the following subsection. This enables the model to iteratively enhance its comprehension and effectively generate precise normalized claims while eliminating the need for extensive fine-tuning.

Our proposed prompt example is shown in Figure 3. Chain-of-thought approaches a complicated problem by efficiently breaking it into a sequence of simpler intermediate stages.

> Identify the central claim in the given post: *Did you know that £20 BILLION a year in tax on Scottish whisky goes straight to England and is spent on their priorities like London's Crossrail, HS2, 2nd Heathrow runway, new Royal Yacht? Did also also know that revenues (£25 BILLION) a year goes to England as well? And that is just the tip of the iceberg. Scotland subsidises England, NOT the other way around!*
>
> Let's think step by step.
>
> The given post claims that the Scottish taxes are sent to England to be spent on their priorities. This claim is verifiable through data statistics. This also has a huge impact on society and seems to contain false information. The claim here needs to be fact-checked. Hence the central claim here is 'Income tax and tax on whisky go to England to be spent on English priorities.'
>
> < ---------------- *more in-context examples here* ---------------- >
>
> Identify the central claim in the given post: **<input post>**
>
> Let's think step by step.

Figure 3: Illustration of our proposed approach. To generate a normalized claim, we use the CACN prompt template, which encompasses explicit task instruction and relevant in-context examples, as well as chain-of-thought reasoning.

**Reverse Check-Worthiness.** The idea about reverse check-worthiness originates from the task of check-worthiness estimation, which in turn is an integral part of the manual fact-checking process (Nakov et al., 2021b). We leverage check-worthiness to steer the model's attention toward salient and pertinent information. By giving the model the ability to produce rationales in natural language that clearly explain the sequence of reasoning stages leading to the solution, we strengthen its capacity for cognitive reasoning with unwavering efficacy. Based on prior research on claim check-worthiness (Barrón-Cedeño et al., 2018; Shaar et al., 2021; Nakov et al., 2022b), we direct our model to prioritize claims that meet specific criteria within the given social media post. These criteria include identifying claims within social media posts that (*i*) contain verifiable factual claims, (*ii*) have a higher likelihood of being false, (*iii*) are of general public interest, (*iv*) are likely to be harmful, and (*v*) are worth fact-checking. For instance, in Figure 3, the claim normalization process begins by identifying the central claim within the input social media post. Subsequently, we reckon the claim's verifiability, i.e., whether it is self-contained and verifiable (e.g., as opposed to not containing a claim or expressing an opinion, etc.). We further evaluate the likelihood of the claim being false and its overall check-worthiness. This step-by-step process ensures a comprehensive analysis of the central claim's characteristics, allowing for effective claim normalization. By incorporating these aspects into our approach, we aim to improve the model's ability to identify and prioritize claims that require scrutiny and verification.

## 5 Experimental Setup

**Baseline Models.** For comparison, we use several state-of-the-art generative systems and categorize them into two groups: (*i*) *Pre-trained Large Language Models (PLMs)*: T5 (Raffel et al., 2020), BART (Lewis et al., 2020), FLAN-T5 (Chung et al., 2022), and PEGASUS (Zhang et al., 2020). For T5, BART, and FLAN-T5, we use base and large model sizes. For PEGASUS, we use the reddit model. (*ii*) *In-context Learning Model:* GPT-3 (text-davinci-003) (Brown et al., 2020).

**Evaluation Measures.** To evaluate lexical overlap, we use ROUGE $(1, 2, L)$ and BLEU-4 (Papineni et al., 2002). We further use METEOR (Banerjee and Lavie, 2005) and BERTScore (Zhang et al., 2019) to assess the similarity between the gold and the generated normalized claims.

**Zero-Shot Learning.** Zero-shot learning aims to apply the previously acquired capabilities of PLMs to similar tasks in a low domain. We hereby assess its suitability for the claim normalization task.

**Few-Shot Learning.** We adopt few-shot learning with 10, 20, 50, and 100 training examples. This gradual exposure to additional labeled data aims to enhance the models' ability to generate accurate and contextually appropriate normalized claims.

**Prompt Tuning.** Prompt-tuning entails adding a specific prefix to the model's input customized to the downstream tasks (Zhang et al., 2023). We investigate the impact of affixing different prompts to the given posts on the performance of T5-based and GPT-3 models. To exert control over the generated normalized claims, we use five control aspects:

tokens, abstractness, number of sentences, claim-centricity, and entity-centricity. A comprehensive description of all these prompts is given in the Appendix (A.2).

**In-Context Learning.** LLMs have the remarkable ability to tackle diverse tasks with a minimal amount of examples given in-context learning prompts (Brown et al., 2020). We use GPT-3 (text-davinci-003) with three different prompts: (*i*) direct prompt (DIRECT), (*ii*) question-guided prompt (Q-GUIDED), and (*iii*) zero-shot chain-of-thought (ZS-CoT). Detailed prompt templates are given in Appendix A.3.

## 6   Experiments and Evaluation

Our experiments reveal that our CACN outperforms all baselines across most evaluation measures. We further examine all systems aiming to answer the research questions listed below.

***Do meticulously crafted prompts enhance the performance of generative models?*** The findings exhibit a significant performance improvement when using prompt-tuning, specifically with in-context examples. Table 3 shows the effectiveness of various prompts across all evaluation measures. However, a notable enhancement of approximately 2–3 points absolute is observed for all semantic measures when transitioning from conventional prompts to our proposed approach when using the same in-context examples. This emphasizes the importance of our framework tailored for the specific task. Moreover, an upsurge in ROUGE-F1 scores (1, 2, and L) emphasizes the resemblance between the generated normalized claims and such created by humans. This, in turn, validates the incorporation of the "*reverse check-worthiness*" chain-of-thought process, which effectively integrates task-specific information into the generative system. We also attempt prompt-tuning in a zero-shot setup; the results are shown in the Appendix (A.2). To summarize, the deliberate design of prompts, along with in-context learning, substantially enhances the performance of generative models.

***Is training models on a specific task less effective than in-context learning with a few examples?*** We observe substantial disparities in the performance of models trained on task-specific data compared to using in-context learning with a limited number of examples, as shown in Table 3.

We can see that the models exposed to in-context examples showcase superior performance, highlighting their efficacy in capturing task-specific patterns. While the trained models exhibit excellence in lexical metrics, their performance in semantic metrics is noticeably lower. Notably, $BART_{LARGE}$, trained on our dataset, outperforms other trained models by sizable margins. These results strongly underline that, within the realm of LLMs, incorporating prompt-tuning with in-context learning holds more promise, leading to enhanced generalization capabilities.

***Do models demonstrate inherent proficiency in generating normalized claims with minimal or no prior training?*** We examine the potential benefits of zero-shot and few-shot learning to investigate the inherent proficiency in generating normalized claims. The zero-shot and the few-shot results are shown in Table 4. Zero-shot learning, which relies solely on the pre-trained language model without any task-specific fine-tuning, performs quite well. On the other hand, few-shot learning does not result in significant improvements. Surprisingly, the models trained using few-shot learning perform slightly worse than zero-shot learning, where the models have no exposure to task-specific data. After training on ten examples, the performance of $FLAN$-$T5_{LARGE}$ drops by 6 BERTScore points absolute, and it continues to decline as more examples are provided.[7] This unexpected result suggests that few-shot learning may be unsuitable for this intricate and complex task. The limited number of examples provided during few-shot learning may have been insufficient for the models to generalize and capture the underlying patterns of normalized claims effectively. Moreover, introducing task-specific data might have introduced conflicting information as these models were never trained on this task, leading to a degradation in performance.

## 7   Qualitative Analysis

**Error Analysis.** To comprehend the performance of CACN, we strive to qualitatively analyze the errors committed by our model in this section. Table 5 shows some randomly selected instances from our test dataset, along with gold normalized claims and predictions from CACN. For comparison, we also show predictions from two best-performing baselines, $BART_{LARGE}$ and DIRECT.

---

[7]See Appendix (A.4) for 50-shot and for 100-shot results.

| | Model | ROUGE-1 | | | ROUGE-2 | | | ROUGE-L | | | BLEU-4 | METEOR | BERTScore |
|---|---|---|---|---|---|---|---|---|---|---|---|---|---|
| | | **P** | **R** | **F1** | **P** | **R** | **F1** | **P** | **R** | **F1** | | | |
| Finetune | T5$_{BASE}$ | 23.65 | 43.60 | 28.99 | 11.38 | 20.70 | 13.90 | 20.45 | 37.91 | 25.15 | 4.57 | 28.86 | 85.13 |
| | T5$_{LARGE}$ | 23.37 | 44.81 | 28.99 | 10.98 | 21.11 | 13.66 | 20.07 | 38.46 | 24.97 | 4.43 | 29.33 | 85.09 |
| | BART$_{BASE}$ | 33.41 | 41.64 | 34.11 | 17.57 | 21.05 | 17.55 | 29.70 | 36.61 | 30.25 | 6.69 | 29.57 | 86.32 |
| | BART$_{LARGE}$ | 35.83 | 42.88 | 36.12 | **19.25** | 21.73 | 18.97 | 31.64 | 37.65 | 31.93 | 7.71 | 31.07 | 86.92 |
| | FLAN-T5$_{BASE}$ | 22.50 | 47.38 | 28.79 | 10.65 | 22.28 | 13.61 | 19.15 | 40.14 | 24.50 | 4.44 | 29.07 | 84.66 |
| | FLAN-T5$_{LARGE}$ | 28.70 | 45.98 | 31.35 | 15.32 | 22.63 | 16.24 | 25.46 | 39.89 | 27.63 | 6.62 | 29.36 | 84.65 |
| In-context | DIRECT | 32.19 | 46.43 | 35.52 | 14.19 | 20.60 | 15.65 | 27.32 | 39.75 | 30.31 | 6.52 | 33.25 | 88.87 |
| | Q-GUIDED | 33.48 | 44.50 | 35.40 | 15.34 | 20.42 | 16.15 | 29.10 | 38.88 | 30.90 | 6.56 | 32.13 | 88.81 |
| | ZS-CoT | 27.77 | 48.39 | 32.42 | 13.18 | 22.33 | 15.16 | 23.84 | 41.51 | 27.79 | 6.37 | 32.93 | 88.44 |
| | CACN (ours) | **37.54** | 46.10 | **38.64** | 18.85 | **23.08** | **19.32** | **33.14** | 40.92 | **34.30** | **9.66** | **35.10** | **89.00** |
| | $\Delta_{CACN-BEST(\%)}$ | ↑4.77 | ↓4.73 | ↑6.98 | ↓2.08 | ↑1.99 | ↑1.85 | ↑4.74 | ↓1.42 | ↑7.42 | ↑25.29 | ↑5.56 | ↑1.15 |

Table 3: Experimental results of CACN and baseline systems on CLAN. We report ROUGE $(1, 2, L)$, BLEU-4, METEOR, and BERTScore. The best scores are shown in **bold**, while the second-best scores are underlined, across each metric. The last row gives the percentage increase in performance between CACN and the best baseline.

| N-shot | Model | ROUGE-1 | | | ROUGE-2 | | | ROUGE-L | | | BLEU-4 | METEOR | BERTScore |
|---|---|---|---|---|---|---|---|---|---|---|---|---|---|
| | | **P** | **R** | **F1** | **P** | **R** | **F1** | **P** | **R** | **F1** | | | |
| 0 | T5$_{BASE}$ | 23.38 | 42.25 | 27.79 | 11.16 | 19.54 | 13.14 | 20.29 | 36.52 | 24.10 | 4.30 | 28.45 | 85.36 |
| | T5$_{LARGE}$ | 24.80 | 43.71 | 29.08 | 12.44 | 20.93 | **14.31** | 21.58 | 37.82 | 25.30 | **5.01** | 29.36 | **85.65** |
| | BART$_{BASE}$ | 22.57 | 47.76 | 27.96 | 11.24 | 22.60 | 13.60 | 19.58 | 40.91 | 24.14 | 4.64 | **30.44** | 85.11 |
| | BART$_{LARGE}$ | 20.80 | 39.68 | 24.41 | 9.86 | 17.20 | 11.16 | 18.23 | 33.85 | 21.20 | 3.92 | 24.67 | 84.78 |
| | FLAN-T5$_{BASE}$ | 30.78 | 32.33 | 28.73 | 15.03 | 15.17 | 13.78 | 28.08 | 28.99 | 26.02 | 4.76 | 23.27 | 83.80 |
| | FLAN-T5$_{LARGE}$ | **31.20** | 34.06 | **30.22** | **15.67** | 16.83 | 15.07 | **28.41** | 30.57 | **27.40** | 5.76 | 25.82 | 84.89 |
| | PEGASUS | 23.96 | 36.07 | 26.16 | 11.69 | 16.76 | 12.48 | 21.14 | 31.50 | 22.99 | 4.87 | 24.87 | 83.07 |
| 10 | T5$_{BASE}$ | 21.55 | 44.22 | 27.44 | 9.87 | 20.36 | 12.57 | 18.31 | 37.77 | 23.40 | 4.04 | 28.64 | 84.97 |
| | T5$_{LARGE}$ | 21.92 | 45.73 | 28.04 | 10.17 | 21.04 | 12.98 | 18.64 | 38.81 | 23.85 | 4.19 | 29.52 | 85.11 |
| | BART$_{BASE}$ | 20.09 | 53.08 | 25.65 | 10.14 | 24.94 | 12.60 | 17.34 | 44.77 | 22.00 | 4.11 | 29.67 | 84.72 |
| | BART$_{LARGE}$ | 19.71 | **53.22** | 25.30 | 9.83 | 24.87 | 12.27 | 17.01 | **45.22** | 21.70 | 4.00 | 29.51 | 84.65 |
| | FLAN-T5$_{BASE}$ | 23.05 | 42.60 | 27.97 | 10.58 | 19.48 | 12.84 | 19.83 | 36.58 | 24.14 | 4.39 | 28.04 | 85.34 |
| | FLAN-T5$_{LARGE}$ | 22.41 | 29.66 | 19.99 | 10.68 | 13.18 | 9.49 | 20.50 | 25.91 | 17.84 | 3.04 | 18.87 | 78.04 |
| | PEGASUS | 15.26 | 40.32 | 20.86 | 7.15 | 18.29 | 9.58 | 13.39 | 35.87 | 18.35 | 3.44 | 24.73 | 80.02 |
| 20 | T5$_{BASE}$ | 21.56 | 44.22 | 27.46 | 9.89 | 20.36 | 12.59 | 18.33 | 37.78 | 23.42 | 4.05 | 28.64 | 84.97 |
| | T5$_{LARGE}$ | 21.97 | 45.80 | 28.07 | 10.24 | 21.11 | 13.03 | 18.69 | 38.77 | 23.86 | 4.21 | 29.46 | 85.12 |
| | BART$_{BASE}$ | 20.08 | 53.16 | 25.63 | 10.14 | **25.03** | 12.60 | 17.34 | 44.89 | 21.99 | 4.11 | 29.63 | 84.72 |
| | BART$_{LARGE}$ | 19.66 | 53.11 | 25.25 | 9.80 | 24.84 | 12.24 | 16.96 | 45.08 | 21.64 | 3.97 | 29.46 | 82.69 |
| | FLAN-T5$_{BASE}$ | 23.15 | 42.80 | 28.11 | 10.63 | 19.58 | 12.91 | 19.93 | 36.76 | 24.27 | 4.44 | 28.20 | 85.34 |
| | FLAN-T5$_{LARGE}$ | 22.38 | 29.79 | 20.00 | 10.69 | 13.25 | 9.51 | 20.48 | 26.00 | 17.85 | 3.04 | 18.96 | 78.01 |
| | PEGASUS | 15.25 | 40.30 | 20.84 | 7.15 | 18.29 | 9.58 | 13.38 | 35.87 | 18.35 | 3.44 | 24.72 | 80.01 |

Table 4: Zero-shot and few-shot performance on our dataset CLAN. We report ROUGE $(1, 2, L)$, BLEU-4, METEOR, and BERTScore.

Naturally, the predictions in the fine-grained analysis are much more intricate than in the coarse-grained quantitative setup. During our manual qualitative analysis, we unveiled several interesting patterns and errors in the generated responses. For example, although BART$_{LARGE}$ generated responses with a high BERTScore in example 1, we noticed that the factual alignment is incorrect, making this model untrustworthy for downstream tasks such as claim check-worthiness and claim verification. In contrast, our proposed model produced a response that is both correct and precise. The response generated by DIRECT is also accurate, but it is excessively long, which contradicts the objective of the normalized claims being concise and straightforward.

This problem is also evident in example 3, where DIRECT produces a factually correct claim but is overly long.

In example 2, we observe that the BART$_{LARGE}$ model demonstrates the lowest number of hallucinations and adheres closely to the input social media post. In contrast, our model's BERTScore performed the worst for this example. However, upon closer inspection, we noticed that the normalized claim that our model generated was indeed correct and most relevant for fact-checking. These findings highlight the complexity and the trade-offs involved in generating normalized claims. While certain models may excel in certain cases, there is often a compromise in other aspects, such as factual accuracy and conciseness.

| | Social Media Post | Normalized Claim | BS |
|---|---|---|---|
| **Sample 1** | They gave you a MASK to cover your face AND then gave you George Floyd's signature phrase 'I can't breath' .… Yet 2.5 years later, most still struggle with the realization that EVERYTHING on TV is fake. Floyd and ConVid never existed | **GOLD:** George Floyd and COVID-19 'never existed'. | – |
| | | **BART:** George Floyd died of COVID-19 | 90.33 |
| | | **DIRECT:** George Floyd's death highlighted the lack of understanding of the reality of events portrayed on television. | 86.84 |
| | | **CACN:** George Floyd and Covid never existed. | 90.98 |
| **Sample 2** | Good News. Finally, CAJY VAZ, an Indian EX Student from St. Xavier's High School from Mumbai, presently based in Goa, found a home remedy for *Covid 19*, which was approved by the WOH for the first time. He proved that *one teaspoon of pepper powder, two teaspoons of honey, a little ginger juice taken for 5 consecutive days can eliminate the effect of corona up to 100%*. The whole world is starting to take this precautionary treatment, finally a happy experience of 2021. | **GOLD:** Honey, Ginger, Black Pepper can cure Coronavirus. | – |
| | | **BART:** One teaspoon of pepper powder, two teaspoons of honey, a little ginger juice taken for 5 consecutive days can eliminate the effect of Covid up to 100%. | 84.89 |
| | | **DIRECT:** Indian ex-student Cajy Vaz found a home remedy of pepper powder, honey and ginger juice that can reduce the effects of Covid-19 up to 100%. | 84.65 |
| | | **CACN:** A home remedy of pepper powder, honey, and ginger juice can eliminate the effect of COVID-19 up to 100%. | 78.47 |
| **Sample 3** | Zelensky sold 17 million hectares of agricultural land to Monsanto, Dupont, and Cargill. Yes, you read it well, 17 million hectares to GM0/chemical companies. This is very bad for the entire world since Ukraine is the largest exporter of wheat and other grains. Zelensky sold 28% of the entire Ukrainian arable land. Australian National Review reports three major US cross-border consortiums have bought 17 million hectares of Ukrainian farmland. To compare: In all of Italy, there are 16.7 million hectares of agricultural land. It turns out that three American companies in Ukraine bought more valuable agricultural land than in all of Italy. The entire area of Ukraine - 600,000 sqm, 170,000 sqm built. | **GOLD:** Ukraine President Volodymyr Zelenskyy sold 17 million hectares of farmland to US corporations. | – |
| | | **BART:** Three American companies have bought 17 million hectares of farmland in Ukraine. | 88.03 |
| | | **DIRECT:** Three major US companies have bought 17 million hectares of Ukrainian agricultural land, which is more than in all of Italy. These companies are Cargill, Dupont and Monsanto, with their main shareholders being American venture capitalists Blackrock, Vanguard and Blackstone. | 83.57 |
| | | **CACN:** Ukrainian President Zelensky sold 17 million hectares of agricultural land to Monsanto, Dupont, and Cargill. | 90.48 |

Table 5: Examples of generated normalized claims along with the gold reference. BART refers to BART$_{LARGE}$.

| Model | Fluency | Coherence | Relevance | Consistency | SC |
|---|---|---|---|---|---|
| **BART** | 3.44 | 3.74 | 3.66 | 3.82 | 3.77 |
| **DIRECT** | 4.48 | 4.58 | 4.03 | 4.26 | 4.38 |
| **CACN** | **4.59** | **4.63** | **4.17** | **4.34** | **4.39** |

Table 6: Human evaluation on the generated normalized claims. SC denotes Self-Contextualized, while BART refers to BART$_{LARGE}$ .

**Human Evaluation.** We conducted an extensive human evaluation to assess the linguistic proficiency of the generated normalized claims. Building upon the measures proposed by van der Lee et al. (2021), we evaluated the generated claims based on four aspects: fluency, coherence, relevance, and factual consistency.[8] We further introduced the parameter of self-contextualization to measure the extent to which the normalized claims included the necessary context for fact-checking within themselves. Each of these measures played a unique and vital role in evaluating the quality of the generated claims. To conduct the evaluation, we randomly selected 50 instances from our test set and assigned five human evaluators to rate every normalized claim on a scale of 1 to 5 for each of these five aspects. All evaluators were fluent English speakers with a Bachelor's or Master's degree. To ensure reliability, each example was evaluated by all five evaluators independently, and then we averaged their scores.

[8]See Appendix A.6 for more detail.

The average scores are presented in Table 6. For comparison, we also included the results from the best-performing baseline systems, namely BART$_{LARGE}$ and DIRECT. Our analysis reveals that the outputs generated by CACN exhibit qualitative superiority compared to the baseline systems across all dimensions.

# 8 Conclusion and Future Work

We introduced the novel task of *claim normalization*, which holds substantial value on multiple fronts. For human fact-checkers, claim normalization is a useful tool that can assist them in effectively removing superfluous texts from subsequent processing. This also benefits downstream tasks such as identifying previously fact-checked claims, estimating claim check-worthiness, etc. We further compiled a dataset of social media posts comprising over 6k posts and their normalized claims. We further benchmarked this dataset with a novel approach, CACN, and showed its superior performance compared to different state-of-the-art generative models across multiple assessment measures. We also documented our data collection process, providing valuable insights for future research in this domain. In future work, we plan to extend the dataset, including with new languages. We also plan to use more powerful LLMs.

## Limitations

While our study has made major contributions to claim normalization, it is critical to recognize and to address its potential limitations. During our data collection process, we excluded claims about images and videos. Yet, we believe that including multimodal information may help improve claim normalization. Another key problem is that each fact-checking organization adheres to its own set of editorial norms, procedures, and subjective interpretations of claims. These variations in writing style and judgments make it challenging to establish a standardized claim normalization. Addressing this issue will necessitate attempts to develop consensus or guidelines among fact-checking organizations in order to ensure greater consistency and coherence in claim normalization. By acknowledging and addressing these limitations, we may endeavor to improve the reliability and soundness of claim normalization systems in the future.

## Ethics and Broader Impact

**Data Bias.** It is important to acknowledge the possibility of biases within our dataset. Our data collection process involves gathering normalized claims from multiple fact-checking sites, each with its own set of editorial norms, procedures, and subjective interpretations. These elements can introduce systemic biases that impact the overall assessment of normalized claims. However, it is important to acknowledge that these biases are beyond our control.

**Environmental Footprint.** Large language models (LLMs) require a substantial amount of energy for training, which can contribute to global warming (Strubell et al., 2019). Our proposed approach, on the other hand, leverages few-shot in-context learning rather than training models from scratch, leading to a lower carbon footprint. It is worth mentioning that using LLMs for inference still consumes a considerable amount of energy, and we are actively seeking to reduce it by using more energy-efficient techniques.

**Broader Impact and Potential Use.** Our model can interest the general public and save time for human fact-checkers. Its applications extend beyond fact-checking to other downstream tasks such as detecting previously fact-checked claims, claim matching, and even estimating claim checkworthiness of new claims.

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

# A Appendix

## A.1 Task Motivation

Claim normalization holds significant promise for combating the spread of misinformation by streamlining fact-checking processes and enhancing the reliability of retrieved evidence. To substantiate our hypothesis regarding the effectiveness of claim normalization, we conducted a well-structured retrieval experiment using the Google API. The objective was to demonstrate the practical benefits of claim normalization in assisting fact-checkers. We randomly selected a sample of 35 instances from our dataset, encompassing social media posts and their normalized claims. Leveraging the capabilities of the Google API, we sought the top-5 most relevant articles for each post and its normalized claim. In a meticulous evaluation process, three annotators individually assessed the relevance (0 or 1) of each retrieved article to the input (post or normalized claim). We then used majority voting to determine the final relevance score for each retrieved article. As depicted in Table 7, the results of our experiment consistently demonstrated the advantage of normalized claims in evidence retrieval. In top-$k$ precision evaluations for various values of $k$ (1, 3, and 5), normalized claims consistently outperformed their corresponding source posts. This observation indicates that claim normalization is not merely a theoretical concept, but significantly enhances the efficiency of evidence retrieval, resulting in more concise and effective tools for aiding the fact-checking process.

| | P@1 | P@3 | P@5 |
|---|---|---|---|
| Post | 0.82 | 0.64 | 0.58 |
| Normalized Claim | **0.88** | **0.73** | **0.69** |

Table 7: Comparative top-$k$ precision evaluations of normalized claim vs. original posts in evidence retrieval.

## A.2 Prompt-Tuning

Raffel et al. (2020) demonstrated that prompt-tuning could enable controllable text generation in T5-based models. We also investigate the impact of affixing different prompts to the given input on the performance of T5-based models for normalized claim generation, along with GPT-3 (text-davinci-003) (Brown et al., 2020). We experimented with various prompts suffixed to the input text before inference, in a zero-shot setting. We discuss our different prompts $P_i$ below.

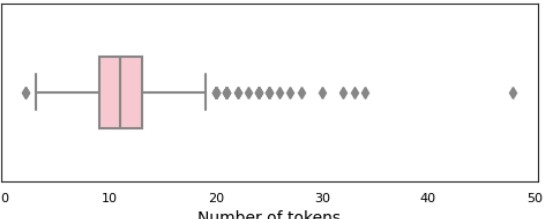

Figure 4: Box-plot for the number of tokens in normalized claims in CLAN.

**Uncontrolled.** We investigated the use of the traditional prompt *'summarize'* for uncontrollable models. This prompt lacks specific control signals, making it *uncontrolled* as it does not provide explicit guidance with specific attributes.

**Token Limit.** We found that normalized claims written by fact-checkers typically adhered to around ten words, as shown in Figure 4. Thus, in order to control the length of the normalized claims, we used the following prompt: "*summarize within the length of 10 tokens*".

**Abstractness.** Abstractness quantifies how much the generated text's words and phrases differ from those extracted directly from it: a fully abstractive summary expresses the central points of the input in very different words and sequences of words compared to the original input. Precisely, the more $n$-grams overlap between a summary and its original document, the less abstractive a summary is. Thus, in order to control the abstractness of the generated normalized claims, we use the prompt "*summarize with abstractness of $a$*", where $a$ represents a value within the range [0;1], denoting the desired level of abstractness. Inspired by Dreyer et al. (2023), we compute the abstractness score $a_i$ for each pair of a post $p_i$ and a normalized claim $s_i$, and then we take an average across all examples:

$$a_i = (1 - X(p_i, s_i)) \qquad (1)$$

where $X$ is the harmonic mean of unigram overlaps precision, bi-gram overlap precision, and longest sub-sequence overlap precision. We found the average abstractness ($a$) to be around 0.8.

**Single Sentence.** The normalized claim written by fact-checkers often consists of a concise single-sentence summary of the post. We use the prompt "*summarise in one sentence*" in order to limit the normalized claims to a single-sentence summary, ensuring brevity and conciseness in delivering the pivotal assertion.

| | Model | ROUGE-1 | | | ROUGE-2 | | | ROUGE-L | | | BLEU-4 | METEOR | BERTScore |
|---|---|---|---|---|---|---|---|---|---|---|---|---|---|
| | | P | R | F1 | P | R | F1 | P | R | F1 | | | |
| T5$_{BASE}$ | UNCONTROLLED | 23.38 | 42.25 | 27.79 | 11.16 | 19.54 | 13.14 | 20.29 | 36.52 | 24.10 | 4.30 | 28.45 | 85.36 |
| | TOKEN-LIMIT | 22.87 | 42.71 | 27.59 | 11.42 | 20.61 | 13.60 | 20.21 | 37.47 | 24.34 | 4.60 | 28.67 | 85.31 |
| | ABSTRACTNESS | 23.02 | 42.21 | 27.58 | 11.40 | 20.20 | 13.47 | 20.17 | 36.79 | 24.16 | 4.53 | 28.46 | 85.41 |
| | SINGLE SENTENCE | 21.06 | 31.19 | 23.07 | 9.61 | 14.02 | 10.46 | 18.30 | 26.75 | 19.96 | 3.57 | 21.77 | 84.41 |
| | CLAIM-CENTRIC | 19.76 | 38.89 | 24.20 | 8.90 | 17.94 | 10.96 | 17.37 | 34.33 | 21.38 | 3.51 | 25.77 | 84.58 |
| | ENTITY-CENTRIC | 19.08 | 32.88 | 22.63 | 8.37 | 14.29 | 9.91 | 16.66 | 28.70 | 19.81 | 3.26 | 20.11 | 83.20 |
| T5$_{LARGE}$ | UNCONTROLLED | 24.80 | 43.71 | **29.08** | **12.44** | 20.93 | 14.31 | 21.58 | 37.82 | **25.30** | **5.01** | 29.36 | 85.65 |
| | TOKEN-LIMIT | 14.51 | 36.24 | 19.53 | 6.24 | 16.01 | 8.45 | 12.79 | 32.47 | 17.33 | 2.56 | 22.13 | 82.08 |
| | ABSTRACTNESS | 23.53 | 44.44 | 28.62 | 11.69 | 20.98 | 13.96 | 20.66 | 38.64 | 25.10 | 5.02 | 29.74 | 85.58 |
| | SINGLE SENTENCE | 23.92 | 44.73 | 28.92 | 11.81 | 21.04 | **14.04** | 20.82 | 38.60 | 25.16 | 4.90 | 29.94 | 85.65 |
| | CLAIM-CENTRIC | 11.61 | 33.36 | 16.44 | 4.95 | 14.77 | 7.08 | 10.50 | 30.70 | 14.97 | 2.38 | 19.64 | 80.99 |
| | ENTITY-CENTRIC | 12.35 | 30.61 | 16.55 | 5.13 | 13.42 | 7.01 | 10.91 | 27.22 | 14.65 | 2.20 | 18.07 | 82.11 |
| GPT-3 | UNCONTROLLED | 19.17 | 48.22 | 24.77 | 7.17 | 19.69 | 9.51 | 16.09 | 40.49 | 20.76 | 3.09 | 27.60 | 86.77 |
| | TOKEN-LIMIT | **26.29** | 40.66 | 28.92 | 9.75 | 15.03 | 10.59 | **22.24** | 34.07 | 24.34 | 3.75 | 26.25 | 87.11 |
| | ABSTRACTNESS | 15.74 | 54.44 | 23.27 | 6.14 | 22.29 | 9.17 | 12.52 | 44.37 | 18.63 | 2.84 | 29.50 | 86.91 |
| | SINGLE SENTENCE | 22.81 | 44.71 | 28.46 | 8.78 | 17.72 | 11.07 | 18.60 | 37.37 | 23.43 | 4.27 | 29.02 | **87.85** |
| | CLAIM-CENTRIC | 14.03 | **56.41** | 21.72 | 5.53 | **23.93** | 8.68 | 11.11 | **46.18** | 17.33 | 2.59 | 25.88 | 86.50 |
| | ENTITY-CENTRIC | 18.92 | 53.52 | 26.34 | 7.74 | 22.41 | 10.82 | 15.26 | 43.90 | 21.34 | 3.67 | **30.98** | 87.16 |

Table 8: Zero-shot prompt-tuning results for T5 and GPT-3 on our datatset CLAN.

**Claim-Centricity.** The task at hand is more than just text summarization; it transcends conventional text summarization by seeking not only to condense the input social media post, but also to discern and to encapsulate the central claim within that input post concisely. Thus, we use the prompt "*summarize the text identifying the central assertion*" to helm the model to focus on the main assertion in the input text.

**Entity-Centricity.** Similarly to the claim-centric prompt, we investigated the technique of creating entity-centered summaries using the prompt, "*summarize the text focusing on the given keywords $(kw_1, kw_2, kw_3, ...kw_n)$*". For this approach, we use Open Information Extraction (Angeli et al., 2015) in order to extract subject–verb–object triples from the input texts.[9] Subsequently, we compile a keyword list encompassing all subjects and all objects within the text. The objective is to direct the model to produce summaries that align well with the subjects and the objects mentioned in the input text.

**Results.** We report the results for our zero-shot prompt-tuning experiments in Table 8. We can see that for the T5-based models, prompt-tuning did not yield any major improvements; rather, it decreased the performance as compared to the *uncontrolled* prompt. However, GPT-3 (`text-davinci-003`) showed some improvements when using these prompts.

### A.3 In-Context Learning Templates

In Figure 5, we show the templates used for the three in-context learning methods used for GPT-3 (`text-davinci-003`) as mentioned in Section 5.

### A.4 Few-Shot Additional Results

We report 50-shot and 100-shot experimental results in Table 9. Interestingly, we observe that introducing more examples to the model did not help it much.

### A.5 Implementation Details

We performed basic data cleaning, e.g., removing non-alphanumeric characters, removing links and hashtags, etc. on our dataset CLAN, using *nltk*. For a standardized evaluation, we relied on widely recognized evaluation libraries such as *py-rouge*,[10] *nltk-bleu*,[11] *nltk-meteor*,[12] and *hugging-face bert-score*.[13] We trained all models for 50 epochs, with early stopping based on validation loss. We set the patience value at 5, and we optimized the models using the Adam optimizer. We set the weight decay to 0.01. For our proposed approach CACN, we used GPT-3 (`text-davinci-0003`) as base model. Finally, we set the maximum length of the generated response to 120 with a temperature of 0.6.

---

[9]https://github.com/philipperemy/Stanford-OpenIE-Python

[10]https://pypi.org/project/py-rouge/
[11]https://www.nltk.org/_modules/nltk/translate/bleu_score.html
[12]https://www.nltk.org/api/nltk.translate.meteor_score.html
[13]https://huggingface.co/spaces/evaluate-metric/bertscore

| Training Samples | Model | ROUGE-1 | | | ROUGE-2 | | | ROUGE-L | | | BLEU-4 | METEOR | BERTScore |
|---|---|---|---|---|---|---|---|---|---|---|---|---|---|
| | | P | R | F1 | P | R | F1 | P | R | F1 | | | |
| 50 | T5$_{BASE}$ | 21.57 | 44.24 | 27.46 | 9.88 | 20.36 | 12.58 | 18.32 | 37.78 | 23.41 | 4.05 | 28.64 | 84.97 |
| | T5$_{LARGE}$ | 22.00 | 45.86 | 28.11 | 10.24 | 21.11 | 13.03 | 18.71 | 38.83 | 23.90 | 4.22 | 29.49 | 85.11 |
| | BART$_{BASE}$ | 20.10 | 53.16 | 25.65 | 10.15 | 25.02 | 12.61 | 17.35 | 44.93 | 22.00 | 4.11 | 29.66 | 84.71 |
| | BART$_{LARGE}$ | 19.63 | 52.96 | 25.20 | 9.78 | 24.73 | 12.21 | 16.94 | 44.88 | 21.61 | 3.97 | 29.46 | 84.65 |
| | FLAN-T5$_{BASE}$ | 23.16 | 42.87 | 28.14 | 10.62 | 19.60 | 12.91 | 19.92 | 36.79 | 24.26 | 4.43 | 28.30 | 85.35 |
| | FLAN-T5$_{LARGE}$ | 22.38 | 29.63 | 19.97 | 10.69 | 13.25 | 9.51 | 20.49 | 25.92 | 17.84 | 3.03 | 18.92 | 77.84 |
| | PEGASUS | 15.24 | 40.28 | 20.83 | 7.15 | 18.29 | 9.58 | 13.38 | 35.84 | 18.34 | 3.44 | 24.71 | 80.02 |
| 100 | T5$_{BASE}$ | 21.55 | 44.24 | 27.45 | 9.87 | 20.36 | 12.57 | 18.31 | 37.78 | 23.40 | 4.04 | 28.65 | 84.97 |
| | T5$_{LARGE}$ | 21.97 | 45.72 | 28.06 | 10.24 | 21.06 | 13.02 | 18.69 | 38.78 | 23.86 | 4.22 | 29.47 | 85.11 |
| | BART$_{BASE}$ | 20.09 | 53.28 | 25.65 | 10.15 | 24.97 | 12.60 | 17.35 | 44.91 | 22.00 | 4.11 | 29.61 | 84.72 |
| | BART$_{LARGE}$ | 19.68 | 53.00 | 25.26 | 9.80 | 24.72 | 12.22 | 16.99 | 44.94 | 21.67 | 3.97 | 29.43 | 84.66 |
| | FLAN-T5$_{BASE}$ | 23.19 | 42.91 | 28.18 | 10.65 | 19.63 | 12.94 | 19.96 | 36.85 | 24.32 | 4.43 | 28.29 | 85.36 |
| | FLAN-T5$_{LARGE}$ | 22.38 | 29.63 | 19.97 | 10.69 | 13.25 | 9.51 | 20.49 | 25.92 | 17.84 | 3.03 | 18.92 | 77.85 |
| | PEGASUS | 15.30 | 40.30 | 20.88 | 7.16 | 18.27 | 9.58 | 13.41 | 35.89 | 18.37 | 3.43 | 24.74 | 80.02 |

Table 9: Few-shot results on our dataset CLAN.

## A.6 Human Evaluation Criteria

Following van der Lee et al. (2021), we define the four human evaluation measures as follows:

1. *Fluency*: measures the linguistic proficiency exhibited by the generated responses;

2. *Coherence*: evaluates the intrinsic structure and the organization of the generated normalized claims;

3. *Relevance*: appraises the discerning selection of contextually appropriate content within the generated response;

4. *Factual consistency*: examines the intricate alignment between the factual accuracy of the generated response and the source text.

**DIRECT**

For the given post get the normalized claim:

Post: "Cough CPR is a form of self CPR. Coughing increases intrathoracic pressure and squeezes blood out of the heart into the aorta. Theoretically, one can remain conscious if one continues to cough during a cardiac arrest (not heart attack without cardiac arrest – all heart attacks do not cause cardiac arrest).
Normalized claim: Cough CPR can prevent cardiac arrest.

*< -------- more in-context examples here -------- >*

For the given post get the normalized claim:

Post: **<input_post>**
Normalized claim: **<generated_response>**

**Q-GUIDED**

What is being claimed in the given post?

Cough CPR is a form of self CPR. Coughing increases intrathoracic pressure and squeezes blood out of the heart into the aorta. Theoretically, one can remain conscious if one continues to cough during a cardiac arrest (not heart attack without cardiac arrest – all heart attacks do not cause cardiac arrest).
The answer is: Cough CPR can prevent cardiac arrest.

*< -------- more in-context examples here -------- >*

What is being claimed in the given post?

**<input_post>**
The answer is: **<generated_response>**

**ZS-CoT**

Identify the central claim in the given post. Let's think step-by-step.

Post: **<input_post>**
Central claim: **<generated_response>**

Figure 5: Our templates for in-context learning prompts used for GPT-3 (`text-davinci-003`).