# OpenReview forum: "$\textit{From Chaos to Clarity}$: Claim Normalization to Empower Fact-Checking"
_EMNLP/2023/Conference — EMNLP 2023 Findings_

### Official Review · Reviewer_H3iM · 2023-08-03

**Soundness:** 4

**Excitement:**

4: Strong: This paper deepens the understanding of some phenomenon or lowers the barriers to an existing research direction.

**Paper Topic And Main Contributions:**

In the era of LLMs, we have lots and lots of new information coming on to the Internet. Some of them are valuable, while others are misinformation. Misinformation can be very harmful and lead to seriously bad outcomes. This paper tackles one angle of this problem, which is claim normalization. This paper tries to extract and simplify a statement in a long noisy social media post, to help readers understand better, and potentially help the downstream classifiers check facts.


**Reasons To Accept:**

The authors curated a dataset CLAN on claim normalization, which if publicly available, could benefit future researchers significantly.

An LLM based approach incorporating chain-of-thought prompting and reverse check-worthiness is also proposed to tackle this research problem. The experiment is thorough and results are reasonable and well support the conclusions.

I really like section 7, both the error analysis section and the human evaluation. In such a new task, I consider quality human evaluation a necessity.

**Reasons To Reject:**

The task itself is interesting, but I am a little concerned about whether this task can benefit future combat of misinformation. That means whether the task of claim normalization can help conquer misinformation or not. The paper mentioned this point in a subjective manner and I am curious if this is indeed true, or an educated hypothesis.


**Reproducibility:**

4: Could mostly reproduce the results, but there may be some variation because of sample variance or minor variations in their interpretation of the protocol or method.

**Reviewer Confidence:**

3: Pretty sure, but there's a chance I missed something. Although I have a good feel for this area in general, I did not carefully check the paper's details, e.g., the math, experimental design, or novelty.

---

> ### Author Rebuttal · Authors · 2023-08-28
>
> **[Task Motivation]** Thanks for your valuable feedback. In essence, claim normalization is a valuable tool that aims to empower fact-checkers by providing them with a refined and consistent framework to navigate the intricacies of misinformation, ultimately contributing to more effective and informed credibility assessments.
>
> We also conducted a simple experiment using the Google API to showcase the effectiveness of claim normalization in assisting fact-checkers. Here's a basic outline of the experiment:
>
> - _Data Collection_: We sampled 35 instances from our dataset comprising social media posts and corresponding normalized claims.
>
> - _Evidence Retrieval_: We leveraged Google API to retrieve the top 5 relevant articles for the post and the normalized claim.
>
> - _Evaluation_: All the retrieved articles were scored 0 or 1 based on their relevance to the given input (post or normalized claim). Three annotators independently labeled every retrieved article, and a majority voting approach was utilized to determine the final relevance score. (With the inclusion of the extra page in the final version, we will be adding this in detail.)
>
> - _Conclusion_: The consistent trend observed across the three setups, where normalized claims excelled in top-k precision evaluations for various k values, demonstrates the utility of claim normalization. This observation suggests that ClaimNorm demonstrates enhanced proficiency in evidence retrieval, making it a more concise and advantageous tool for aiding fact-checking procedures. Furthermore, the discernible performance drop in post-based retrieval, mainly as k increased from 1 to 3, highlights the robustness of ClaimNorm's approach in maintaining factual precision and minimizing complications.
>
> | Input query      | Precision@1  | Precision@3  | Precision@5  |
> |------------------|------|------|------|
> | Post             | 0.82 | 0.64 | 0.58 |
> | Normalized Claim | **0.88** | **0.73** | **0.69** |

---

### Official Review · Reviewer_uYfo · 2023-08-05

**Soundness:** 3

**Excitement:**

2: Mediocre: This paper makes marginal contributions (vs non-contemporaneous work), so I would rather not see it in the conference.

**Paper Topic And Main Contributions:**

This paper proposes claim normalization, a new task that aims to decompose complex social media posts into normalized claims. This paper then proposes CACN, an LLM-based approach towards this new task which has great performance on the proposed dataset.

**Reasons To Accept:**

+ claim normalization is an interesting task
+ understanding and processing social media texts is an important research question

**Reasons To Reject:**

1. I strongly recommend that the authors validate the quality of the GFC API output to ensure the accuracy of the ground truth. This step is crucial in ensuring the reliability of the findings.
2. Since the ground truth output can be assess on the Internet, which may lead to test data leakage problem. I encourage authors to discuss the data leakage problem and its implications in this work.
3. The novelty of the proposed CACN method appears limited, as it seems to be a direct combination of CoT and Reverse check worthiness. Additionally, the claim normalization task could be seen as a form of text augmentation or summarization.
4. The description of the baseline is not sufficiently clear. For instance, it is unclear whether the finetuning refers to full-parameter or parameter-efficient prompt tuning. Clarifying this aspect would enhance the understanding of the experimental setup.
5. It is essential to ensure fairness in the comparisons made. For example, the same method should be applied to different models, and the same base model should be evaluated with different prompt paradigms. This would provide a more comprehensive and unbiased analysis.
6. Table 4 lacks results for the proposed CACN method. It is important to include these results to provide a complete evaluation of the proposed approach under 0 and few shot setting.
7. The performance degradation in few-shot learning compared to 0-shot learning requires further explanation. I recommend that the authors examine the few-shot learning exemplars and provide additional insights in this aspect.

**Reproducibility:**

3: Could reproduce the results with some difficulty. The settings of parameters are underspecified or subjectively determined; the training/evaluation data are not widely available.

**Reviewer Confidence:**

4: Quite sure. I tried to check the important points carefully. It's unlikely, though conceivable, that I missed something that should affect my ratings.

---

> ### Author Rebuttal · Authors · 2023-08-28
>
> Thank you for your input; we value the opportunity to address your concerns.
>
> **[Quality of the GFC API Output]** We appreciate the recommendation for quality validation of the GFC API. We would like to mention that the GFC API primarily functions as a platform for listing fact-checked posts; it compiles claims that have already undergone fact-checking by credible fact-checkers and journalists. Given its role, the GFC API's reliability is inherently high as it aggregates content produced by trusted sources. Furthermore, we included fact-checking sites with recognition from the International Fact-Checking Network (IFCN) to enhance credibility.
>
> **[Data Leakage]** We appreciate your concern and acknowledge the significance of discussing this matter explicitly within our work. Acknowledging the significant concern of data leakage in models like GPT, we have conducted a small experiment to assess its impact on our approach. We have evaluated our proposed model's performance using recent posts from the past two months (given GPT-3's training up to September 2021). Encouragingly, the results highlight that our model excelled even with new posts. This reaffirms the robustness of our approach amidst potential data leakage challenges.
>
> |           | ROUGE-1 |       |       | ROUGE-2 |       |       | ROUGE-L |       |       | BLEU  | METEOR | BERTScore |
> |------------------|---------|-------|-------|---------|-------|-------|---------|-------|-------|-------|--------|-----------|
> | Models           | P       | R     | F     | P       | R     | F     | P       | R     | F     |       |        |           |
> | CACN (Test set)    |   37.54 |  46.1 | 38.64 |   18.85 | 23.08 | 19.32 |   33.14 | 40.92 |  34.3 |  9.66 |   35.1 |        89 |
> | CACN (In-the-wild) |   38.74 | 51.59 | 42.07 |    24.6 | 32.55 |  26.7 |   36.82 | 49.59 | 40.29 | 16.25 |   38.9 |     87.05 |
>
> In our revised version, we will include a comprehensive section addressing the data leakage problem and its potential impact.
>
> **[Novellty of the Task]** The claim normalization task shares similarities with text augmentation or summarization. However, it is important to note that the purpose of claim normalization goes beyond simple augmentation. It involves identifying relevant information, reducing ambiguity, and ensuring factual consistency, ultimately improving the credibility assessment process. This task's application within the broader context of fact-checking contributes to easing the segregation of claims (misinformation) within noisy social media posts. In claim normalization, the focus is on extracting the core claim or statement within the text. In contrast, text summarization condenses the entire content while retaining essential details.
>
> **[Baselines]** We have fine-tuned the model weights for our downstream task using our task-specific dataset (CLAN). The objective is to optimize the model's performance for the particular task while retaining the general language understanding and generation capabilities it gained during pre-training. With the inclusion of the extra page in the final version, we will be adding these details.
>
> **[System Comparison]** To perform unbiased results, we conducted experiments under consistent conditions to ensure an unbiased analysis. We employed identical prompts and in-context examples across all models, allowing for direct and meaningful comparisons. Furthermore, we maintained consistency by employing identical examples for fine-tuning across various models, specifically in the context of diverse few-shot setups.
>
> **[Inclusion of results]** We appreciate your suggestion and will ensure its incorporation.
>
> **[Degradation in Few-shot Setting]** We acknowledge the significance of addressing this thoroughly. The task is new and challenging; the models haven't encountered such a situation before. The shift from zero-shot to few-shot learning might perplex the models, resulting in a performance dip. This also sheds light on the intricacies inherent in the task, highlighting the complex relationships between the data points and the model's understanding. This insight is a valuable stepping stone in comprehending the dynamics of few-shot learning and its application in novel scenarios.

---

### Official Review · Reviewer_GSfB · 2023-08-12

**Soundness:** 4

**Excitement:**

4: Strong: This paper deepens the understanding of some phenomenon or lowers the barriers to an existing research direction.

**Paper Topic And Main Contributions:**

The authors address noise within social media posts that may hinder proper identification of the main claim or assertion of post. As such, this paper proposes a prompt-tuning method (CACN) which aims to eliminate noise within a post, resulting in a "normalized claim” more suitable for the verification of an assertion. This paper offers two primary contributions: firstly, the foundational methodology of CACN, and secondly, the development of a novel dataset (CLAN) that includes fact-checked social media posts paired with their respective normalized claims, which can be used for downstream tasks.

**Reasons To Accept:**

The proposed approach behind CACN is well-justified and novel. Much thought has been put into given prior research to better the subcomponents of CACN.

The procedure for creating CLAN is clear to understand and replicate. Furthermore, the performance of the usage of CACN is evaluated in depth using multiple metrics and models. There has been good effort put to ensure robustness.

The paper overall is written with great clarity, flow, and cohesion. Nearly every section was easy to understand.

**Reasons To Reject:**

My main concerns surround the usage of GPT models. There is reason to believe a contamination issue may exist. As a closed model, we do not know what data the model has trained on. As such, when using the Chain-of-thought Prompting, the model may conceivably be using the information it is aware of regarding the given assertion to influence the step-by-step procedure which then identifies the main claim. This may not seem as much of a problem, but let's use one of the sample posts given as an example:

Social Media Post: Turmeric and lemon are two simple, cheap and handy things you can use regularly to fight #CoronaVirus. Homemade rasam is also very useful.

Normalized Claim: Turmeric and lemon help fight against coronavirus.

Let’s consider why the GPT model chose to focus on “Turmeric and Lemon” as the primary assertion and not “Homemade rasam”. While it's logical to see rasam as secondary, the model might lean towards "Turmeric and Lemon" due to its training exposure to these terms in health contexts. This familiarity, combined with the "Step-by-Step" approach of Chain-of-thought Prompting, might skew prioritization. Such biases could overshadow lesser-known or context-specific details in the original post, affecting the claim's genuineness. More commonly discussed or popularized topics may be given more precedence over lesser-known or more specific aspects of the original post, regardless of the actual validity or relevance.

As another example, take another one of the sample posts:

Social Media Posts: Certain meats, like beef and chicken, contain an amino acid called leucine. Leucine turns on the chain reaction that stimulates the skin oil’s glands and makes the acne breakouts more likely.

Normalized Claim: Acne breakouts are more common in non-vegetarians.

The original doesn't mention vegetarianism, yet the normalized claim infers it. Quite reasonably, a claim could be drawn regarding “Certain meats, like beef and chicken, contain an amino acid called leucine” or that “Leucine turns on the chain reaction that stimulates the skin oil’s glands and makes the acne breakouts more likely” instead. Clearly, the model used pre-existing knowledge regarding acne in the context of vegetarianism to draw a connection to phrase it as such. Different interpretations might be factually accurate, but they aren't the same claims.

These examples illustrate potential biases in using LLMs for claim normalization. Although Table 3 may show CACN's efficiency, the provided examples from the sample data suggest that the normalized claim, while valid, might deviate in meaning from the original. As per my understanding, since there is no way presented to prevent such occurrences, it may be problematic.

Although human evaluation is done to evaluate fluency, coherence, relevance, and factual consistency, it is only evaluated on 20 out of the 6338 posts (.3%). This seems too low of a sample size to draw claims of “qualitative superiority”.

**Reproducibility:**

5: Could easily reproduce the results.

**Reviewer Confidence:**

4: Quite sure. I tried to check the important points carefully. It's unlikely, though conceivable, that I missed something that should affect my ratings.

---

> ### Author Rebuttal · Authors · 2023-08-28
>
> Thanks for the valuable feedback.
>
> **[Usage of GPT Models]** We agree that GPT models are closed, meaning we lack direct access to all their training data. This opacity raises concerns about potential biases and misinformation in the model. However, these models are pre-trained on diverse datasets to reduce bias risk. Further, to address this, we're taking a multi-step approach. We begin with a pre-trained GPT model, then fine-tune it for our specific task, aligning outputs with our needs. The vast training data's advantage is two-sided: enhancing capabilities but demanding careful monitoring for issues. By meticulously crafting the prompt and monitoring, we aim to benefit from the data while reducing contamination risks.
>
>
> **[GPT Data Contamination]** Thank you for highlighting this critical concern. Since GPT-3 was trained on data available up to September 2021, we meticulously sampled 20 recent fact-checked social media posts (July-August 2023) and leveraged our model to generate normalized claims. Impressively, these recent claims demonstrated comparable effectiveness, reaffirming the relevance of our approach. This highlights the method's adaptability and high resistance to data leakage concerns, strengthening our commitment to real-world applicability and credibility. Further, the slightly amplified scores are attributed to the smaller sample size in our in-the-wild analysis.
>
> | Models           | ROUGE-1       | | | ROUGE-2 |       |       | ROUGE-L |       |       | BLEU  | METEOR | BERTScore |
> |------------------|---------|-------|-------|---------|-------|-------|---------|-------|-------|-------|--------|-----------|
> |                  | P       | R     | F     | P       | R     | F     | P       | R     | F     |       |        |           |
> | CACN (Test set)    |   37.54 |  46.1 | 38.64 |   18.85 | 23.08 | 19.32 |   33.14 | 40.92 |  34.3 |  9.66 |   35.1 |        89 |
> | CACN (In-the-wild) |   38.74 | 51.59 | 42.07 |    24.6 | 32.55 |  26.7 |   36.82 | 49.59 | 40.29 | 16.25 |   38.9 |     87.05 |
>
> **[Extended Human Evaluation]** We appreciate your feedback and have undertaken an extended human evaluation on more test samples under the same setup reported previously. The results we gathered comprise 50 samples and consistently align with the findings initially presented.
>
> |        | Fluency | Coherency | Relevance |   Factual Consistency |   Self-Contextualised |
> |--------|--------:|----------:|----------:|-----:|-----:|
> |   BART |    3.44 |      3.74 |      3.66 | 3.82 | 3.77 |
> | DIRECT |    4.48 |      4.58 |      4.03 | 4.26 | 4.38 |
> |   CACN |    4.59 |      4.63 |      4.17 | 4.34 | 4.39 |

---

### Meta-Review · Area_Chair_BEVm · 2023-09-18

**Recommendation:** 4

**Metareview:**

The paper introducing CACN offers a well-justified and innovative approach that builds upon prior research. The method for developing CLAN is transparent and easily replicable. There is commendable depth in evaluating CACN's performance, ensuring robustness using various metrics and models. The manuscript is praised for its clarity, flow, and cohesion. The topic of claim normalization, especially in the context of social media texts, is seen as both intriguing and significant. A notable highlight is the curated dataset, CLAN, which has potential value for future research if made public. The paper's LLM-based approach, with chain-of-thought prompting and reverse check-worthiness, is robust with well-supported results. Section 7, especially the error analysis and human evaluation, stands out for its quality. It is recommended that the authors consider feedback to further refine their contributions.

---

### Decision · Program_Chairs · 2023-10-07

**Decision:**

Accept-Findings

**Comment:**

The paper introducing CACN offers a well-justified and innovative approach that builds upon prior research. The method for developing CLAN is transparent and easily replicable. There is commendable depth in evaluating CACN's performance, ensuring robustness using various metrics and models. The manuscript is praised for its clarity, flow, and cohesion. The topic of claim normalization, especially in the context of social media texts, is seen as both intriguing and significant. A notable highlight is the curated dataset, CLAN, which has potential value for future research if made public. The paper's LLM-based approach, with chain-of-thought prompting and reverse check-worthiness, is robust with well-supported results. Section 7, especially the error analysis and human evaluation, stands out for its quality. It is recommended that the authors consider feedback to further refine their contributions.